# Imaging the electrical activity of organelles in living cells

Ella Matamala[1,2], Cristian Castillo[2], Juan P. Vivar[1], Patricio A. Rojas[3] & Sebastian E. Brauchi ⓘ [1,2,4 ✉]

Eukaryotic cells are complex systems compartmentalized in membrane-bound organelles. Visualization of organellar electrical activity in living cells requires both a suitable reporter and non-invasive imaging at high spatiotemporal resolution. Here we present hVoS$_{org}$, an optical method to monitor changes in the membrane potential of subcellular membranes. This method takes advantage of a FRET pair consisting of a membrane-bound voltage-insensitive fluorescent donor and a non-fluorescent voltage-dependent acceptor that rapidly moves across the membrane in response to changes in polarity. Compared to the currently available techniques, hVoS$_{org}$ has advantages including simple and precise subcellular targeting, the ability to record from individual organelles, and the potential for optical multiplexing of organellar activity.

[1] Physiology Institute, Universidad Austral de Chile, Valdivia, Chile. [2] Millennium Nucleus of Ion Channel-Associated Diseases (MiNICAD), Valdivia, Chile. [3] Laboratory of Neuroscience, Faculty of Chemistry and Biology, Universidad de Santiago de Chile, Santiago, Chile. [4] Janelia Research Campus, Howard Hughes Medical Institute, Ashburn, VA, US. ✉email: sbrauchi@uach.cl

In general, the regulated flow of ions and charge separation establishes a voltage gradient across semipermeable membranes. The voltage gradient across organelle membranes ($\Psi_{org}$) will be then defined by the specific set of ion channels and transporter proteins expressed on a given organelle. $\Psi_{org}$ is modulated by intracellular signaling cascades and is likely to be essential to the maintenance of organellar homeostasis[1–6]. Although the central importance of the electrical activity of organelles has been widely acknowledged, the detailed mechanisms that support this type of signaling are poorly understood, partially owing to the lack of sophisticated research tools[7–11].

Membrane potential imaging using voltage-sensitive dyes has been extensively used for mitochondria and endoplasmic reticulum (ER); and more recently, for phagosomes and lysosomes[12–16]. Still, a precise and standardized method allowing for the recording of electrical signals generated at individual organelles is not yet available. To this end, we have developed a general methodology for the recording of electrical signals from single organelles in living cells. The method relies on the use of a Hybrid Voltage Sensors (hVoS)[17] and here we show its effectiveness for the fast imaging of variations in $\Psi_{org}$ ($\Delta\Psi_{org}$). The hVoS approach is extremely sensitive, capable of measuring rapid changes in the membrane potential of both excitable and non-excitable cells, under single- or multi- photon excitation imaging[17–21]. The method takes advantage of a fluorescence resonance energy transfer (FRET) pair consisting of a membrane-anchored fluorescent protein acting as donor and the non-fluorescent hydrophobic anion dipicrylamine (hexanitrodiphenylamine; DPA) acting as acceptor. Owing to its small size, negatively charged DPA has the ability to rapidly transit across the dielectric in response to changes in the membrane potential, acting as voltage sensor[17,22]. Conveniently, hVoS readout of membrane potentials showed a broad dynamic voltage range (i.e., $-100$ to $+30$ mV)[19,21,23].

Lysosomes are degradative organelles essential to maintain cellular metabolic activity. Their direct association with mTOR kinases is thought to integrate their catabolic role with different signaling cascades in the cell, including ion channel activity[24]. Several channels, transporters, and ion pumps such as the vesicular proton pump (v-ATPase), Two-Pore $Na^+$ Channels (TPCs), TMEM175 $K^+$ channels, calcium-activated BK channels, members of the mucolipin subfamily of TRP channels (TRPMLs), SLC, CLC, and CLIC transporters, have been described as active residents of the endolysosomal system[24–27]. The expression patterns and localization of these proteins combined with electrophysiological data have led to the proposal that the lysosome is an electrically active organelle[6,15].

Optical recordings of the membrane potential in lysosomal membrane ($\Psi_{ly}$) have been previously accomplished by using a combination of potentiometric fluorescent dyes (i.e., oxonol derivatives) forming a FRET pair with fluorophores that are preferentially segregated to the lysosome membrane[15]. The high density and variety of organelles within the endolysosomal system impose restrictions on both resolution and targeting. Thus, the isolation of the individual contributions from endosomes, lysosomes, or phagosomes is not possible using such approach.

Here, we focused our attention on lysosomes because their well-described electrical response to mTOR inhibitors (e.g., rapamicyn), which provide a suitable pharmacological tool to induce and observe variations in $\Psi_{ly}$[6]. To optically follow such changes in $\Psi_{ly}$, we recorded voltage-sensitive FRET signals between DPA and a fluorescent protein of choice (XFP) fused to the cytoplasmatic C-terminal domain of the lysosomal-associated membrane protein 1 (Lamp1). Our results indicated that hVoS$_{org}$ reliably reports the amplitude of $\Psi_{ly}$ and kinetics of $\Delta\Psi_{ly}$ at the level of single organelle. Being a single wavelength FRET tool, based on cellular markers of common use, the versatility of the technique allowed us for out-of-the-box recordings of other intracellular compartments. Therefore, in addition, we report the absolute resting potential of Golgi and ER membranes in four different cell types as examples of both whether the technique could be easily expanded and its limitations.

## Results

**Characterization and targeting of hVoS$_{org}$ to lysosomal membranes.** In principle, when combined with DPA, it is possible for many membrane-bound fluorescent markers to transduce voltage changes occurring at the target membrane into fluorescence fluctuations (Fig. 1a, b). Therefore, fluorescently tagged organelle markers provide a handy tool to selectively image the electrical activity of internal membranes. For hVoS$_{org}$ to work, DPA must reach intracellular membranes (Fig. 1c). Once in a membrane, it would distribute according to a voltage-dependent equilibrium[17,21,22]. In such a scenario, each individual membrane compartment of the cell would define three equilibriums governing the distribution of DPA molecules; lipid–water interphase at each side of the membrane and a voltage-dependent barrier governing DPA transit between them (Fig. 1c). To first confirm that DPA can reach organellar membranes in living cells, we used two fluorescent markers simultaneously. A farsenylated EGFP ($_f$EGFP) used in previous studies to report changes in plasma membrane potential was used in combination with Lamp1-mCherry that targets the lysosome (Fig. 1d, e). Both fluorescent signals quenched upon the addition of DPA (4 µM) (Fig. 1f), confirming the ability of DPA to reach internal membranes in living cells and the effectiveness of the intracellular FRET pair.

According to previous studies, the efficiency of hVoS is larger for fluorescent proteins that are excited at lower wavelengths[19]. Moreover, a complete framework detailing voltage-dependent quenching of membrane-bound EGFP by DPA is available[19,21,22]. Thus, to take advantage of the available parameters calculated for the FRET pair EGFP/DPA, all quantitative optical measurements of $\Psi_{org}$ in this work were performed using EGFP-fused markers. Accordingly, to evaluate the distribution of DPA across cellular membranes, we recorded quenching of fluorescence in HEK293 cells expressing either $_f$EGFP or Lamp1-EGFP. Our model suggests that although the system would be at equilibrium in ~900 seconds, DPA will reach 95% saturation (~1.5 mM) at the membrane in ~500 seconds (Supplementary Data 2). The model estimates that the final concentration of DPA once at equilibrium would be about 2 µM and 30 µM at the cytoplasm and lysosomal lumen, respectively, and 1.9 mM and 0.3 mM at the cytoplasm and lysosomal membrane, respectively, (Supplementary Data 2).

Our lamp1-XFP constructs display a characteristic distribution that space-correlates well with other lysosome-resident proteins. This is the case of the ion channel proteins mucolipin 1 (TRPML1) and two-pore sodium channel 1 (TPC1) (Fig. 2a; Pearson's coefficients of $0.90 \pm 0.01$ and $0.85 \pm 0.063$, respectively). In contrast, the ER marker Sec61b and the Golgi marker ManIIa do not colocalize well with Lamp1 (Supplementary Fig. 1; Pearson's coefficients of $0.45 \pm 0.09$ and $0.48 \pm 0.06$, respectively). Epifluorescence images revealed a characteristic ring shape of the Lamp1-EGFP positive structures (Fig. 2b). The ring-shaped objects were usually between 1.2 and 0.9 µm, consistent with the dimensions of lysosomes (Fig. 2c). Lysosomes are moving objects inside the living cell; therefore, we evaluated the relative mobility of Lamp1-EGFP endomembranes[28]. By computing mobility maps, we determine that at 20°C Lamp1-EGFP positive

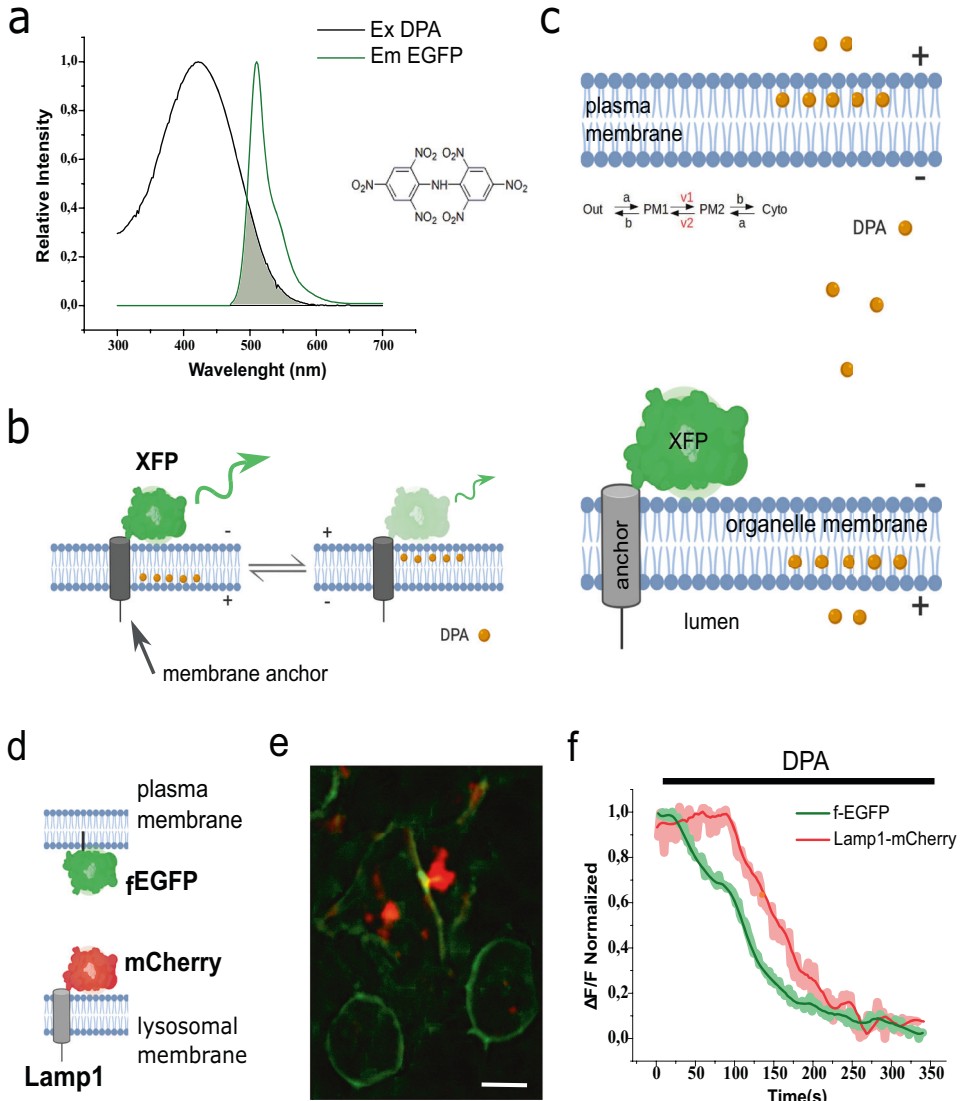

**Fig. 1 Dipicrylamine (DPA) reaches internal membranes. a** Overlap between the spectra corresponding to EGFP emission and DPA absorption. A DPA molecule is pictured as inset. **b** Schematic representation of DPA transit in the HVoS technique. The negatively charged DPA accumulates preferentially at the positive leaflet of the membrane and redistributes in response to changes in membrane voltage. Redistribution of DPA within the membrane cause distance-dependent quenching of EGFP emission. **c** Schematic representation of DPA incorporation and distribution within the cell. Equilibrium describes the transit of DPA within a single membrane. Once at the membrane, DPA distributes according to a voltage-dependent equilibrium with voltage-dependent rates v1 and v2. **d** Cartoon depicting the topology of the probes used for simultaneous measurements of plasma membrane and lysosomes. **e** Simultaneous expression of Lamp1-mCherry (lysosomal marker) and farnesylated GFP (fEGFP; plasma membrane marker) in HEK293 cells. Bar = 10 μm) **f** Time course of fluorescence for the cell in panel **e** during DPA incorporation. The lysosomal signal (red trace) corresponds to the average of six spots proximal to the membrane and centered in the cell. The plasma membrane signal (green trace) was extracted from four regions of equal size.

structures are relatively immobile during the two-minute window required for our recordings, eliminating the need for further correction of motion (Fig. 2d).

In hVoS$_{org}$, the voltage sensor (i.e., DPA) is a small molecule that moves closer or away from a membrane-anchored XFP (Fig. 1b). Therefore, the precise topology of the fluorescent marker is critical for the correct interpretation of the FRET readout in terms of membrane polarity. To confirm the topology of Lamp1-EGFP (Fig. 3a), we performed a fluorescence protease protection (FPP) assay[29]. In this procedure, a pulse of digitonin (5 μM) that permeabilizes the plasma membrane is followed by rinsing and further incubation with trypsin (4 mM). By doing this, the final protease digestion eliminates the signal coming from XFPs facing the cytoplasm while luminal XFPs remain protected (Fig. 3b). In agreement with our design, we observed

that Lamp1-EGFP fluorescence is rapidly quenched by trypsin (Fig. 3c, Top). A second lysosomal marker was used as control. The lysosome-targeted probe Lyso-pHoenix places the green-emitting pH sensor pHluorin at the luminal side and the red-emitting fluorescent protein mKate facing the cytoplasm[30]. In agreement with the reported topology, we observed that pHluorin signal remains stable while mKate fluorescence quenches upon the addition trypsin (Fig. 3c, Bottom).

**hVoS$_{org}$ signals are sensitive to changes in membrane voltage.** An important contributor to the lysosomal membrane potential is the pH gradient (ΔμH$^+$), maintained by the vesicular proton pump (v-ATPase)[31]. Thus, perturbations of ΔμH$^+$ would provide a simple approach to test our ability to measure changes in Ψ$_{ly}$ in

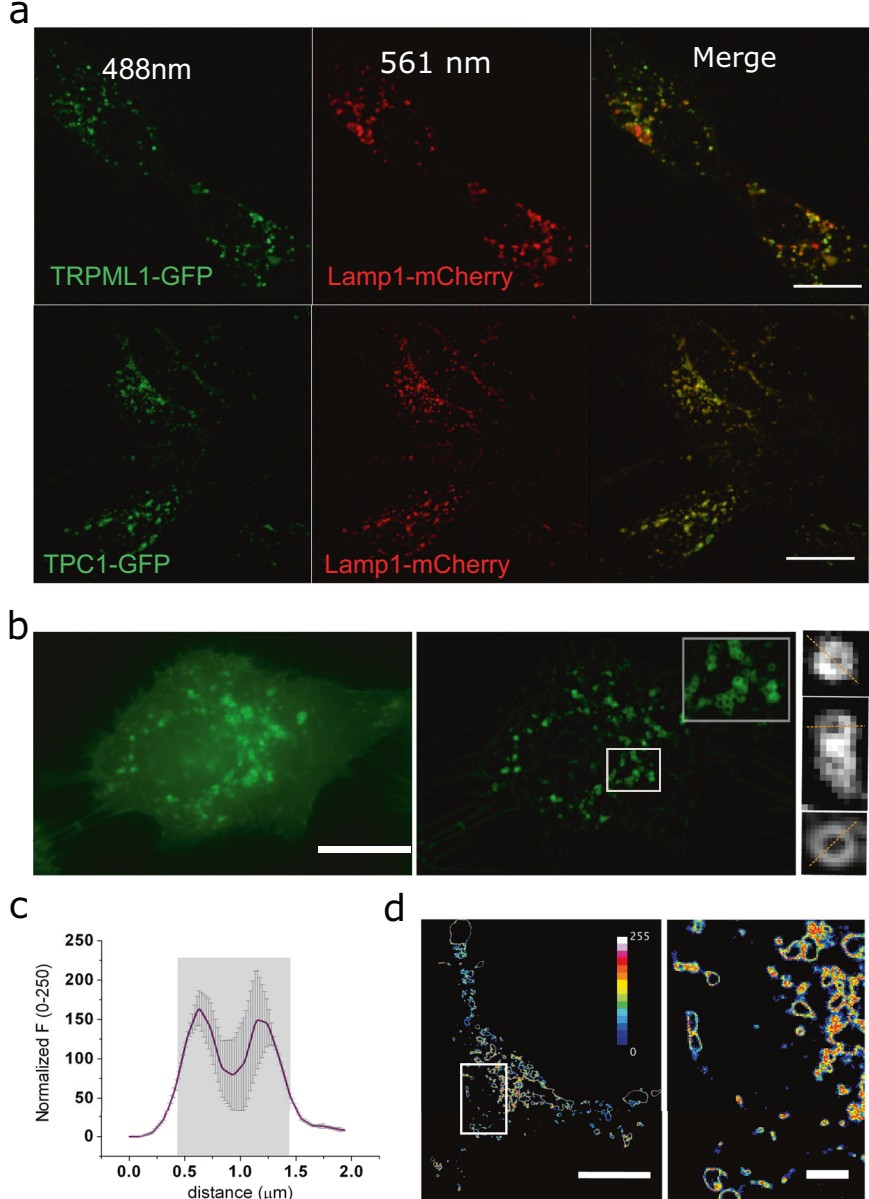

**Fig. 2 Imaging of subcellular structures identified as lysosomes. a** Representative colocalization images of the lysosomal marker Lamp1 with the lysosomal ion channels TPC1-EGFP and TRPML1-EGFP. Scale bars correspond to 5 μm. **b** Raw image of a transiently transfected HEK293 cell expressing Lamp1-EGFP (left). FFT filtered and background-subtracted image (middle) allows the identification of individual lysosomes. Inset corresponds to a 6 × 6 μm zoom of the highlighted area. An average of 10 frames allows obtaining better resolved individual lysosome structures (right panel). The orange lines indicate the axis used to extract the intensity profile in Lamp1-positive objects. **c** Intensity versus distance plot showing the average size of lysosomes in the study. Gray values were collected along orange lines indicated in **b**. The average size of the doughnut-shaped objects was 112 ± 33 nm ($n = 26$). error bars correspond to SEM. **d** Mobility map of a cell expressing Lamp-EGFP (left; Scale bar, 10 μm). The white square indicates the region zoomed on the right panel (Scale bar, 2 μm). Red shades indicate zones with high motion and black represent regions containing immobile objects.

living cells. Following previous reports[15,32], we first induced alkalinization of the lysosomal lumen by adding ammonium to the extracellular solution. It has been estimated that 20 mM ammonium in the extracellular solution will depolarize the lysosomal membrane in ~40 mV[15]. As expected, ammonium incubation (10 mM) causes a rapid and strong change in luminal pH, observed as an increase of the fluorescence signal measured by the pH sensor pHluorin, which we localized to the lysosomal lumen by transiently expressing Lyso-pHluorin[30] (Fig. 3d, e). Next, we repeated the experiment using Lamp1-EGFP in the presence and absence of DPA. In the presence of DPA, ammonium treatment quenches about 20% of EGFP's fluorescence, indicating a voltage-dependent transit of DPA molecules across the lysosomal membrane (Fig. 3d, e). Such quenching is absent when the DPA is not present (Fig. 3e). On the contrary, a modest increase in EGFP's fluorescence can be detected immediately after the ammonium treatment in the absence of DPA (Fig. 3e). This could be explained because $NH_4^+$ will alkalinize not only the lumen of organelles but also the cytoplasm. It has been reported that 20 mM ammonium in the external solution will cause a change in cytoplasmic pH of ~0.3 pH units[33]. Given the high buffer capacity of the cellular cytoplasm, we reasoned that normal fluctuations in cytoplasmic pH would not be sufficient to contaminate membrane potential measurements performed with $hVoS_{org}$.

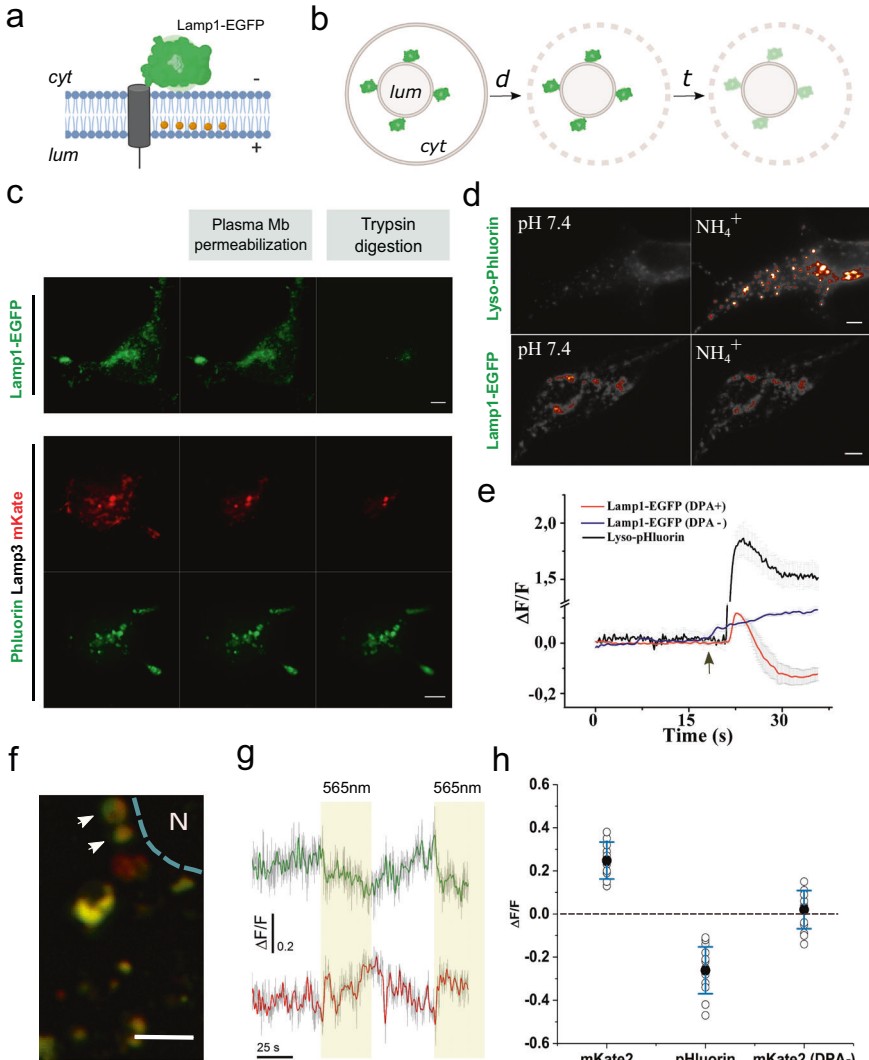

**Fig. 3 Topology of Lamp1-EGFP and the effect of lysosomal pH on hVoS_org signals. a** Cartoon of the predicted topology of the FRET pair with EGFP facing the cytoplasm and DPA accumulated at the luminal leaflet of the lysosomal membrane. **b** Schematic representation of the FPP assay illustrating the position of the fluorescent tags relative the organelle membrane. A 2-min treatment with 10 μM digitonin (d) is followed by perfusion with ringer containing 4 mM trypsin (t). **c** Lamp1-EGFP exposes the fluorescent protein to the cytoplasm as revealed by protease protection assay (top) (scale bar, 5 μm). Representative image of HEK-293T cells expressing lyso-pHoenix (pHluorin-CD63-pHluorin-Arch3-mKate) were subjected to the FPP assay (bottom panels) (scale bar, 5 μm). **d** Representative images of HEK293 cells expressing the pH sensor pHluorin (top) or EGFP (bottom). Images were taken before (left) and after (right) ammonium treatment. **e** Time course of fractional fluorescence during ammonium treatments. The arrow denotes addition of NH4+. **f** Representative image of individual lysosomes expressing Lyso-pHoenix. Arrows indicate representative individual lysosomes. N indicates the position of the nucleus. (scale bar, 3 μm). **g** Traces of pH signal (green shades) and membrane voltage signal (red shades) acquired in response to light stimulation (indicated in yellow). Traces were obtained by averaging the response of the lysosomes indicated in **f**. Solid lines correspond to the smoothened signal. **h** Average values of ΔF/F for the change in membrane voltage (mKate2), luminal pH (pHluorin) ($n = 5$ independent paired experiments), and the voltage-insensitive mKate2 in the absence of the voltage sensor DPA ($n = 3$ independent experiments). Error bars indicate mean ± SD.

Under our imaging conditions and according to the topology of the probe (i.e., XFP facing the cytosol), a negative deflection of the fluorescence signal is caused by the redistribution of DPA molecules to the outer leaflet of the lysosomal membrane, and we interpret this change as depolarization. According to this, our results indicate that alkalinization of the lysosome lumen causes a rapid depolarization of the organelle's membrane as reported before ref. [15].

To test the ability of hVoS_org system to follow rapid changes in $\Psi_{ly}$ ($\Delta\Psi_{ly}$), we took advantage of available optogenetic tools. *Lyso-pHoenix* is a large protein sensor composed of mKate at the cytoplasmic N-terminal domain followed by the light activated proton pump Arch, and pHluorin on the luminal C-terminal domain of the protein[30]. The sensor targets the lysosome via a Lamp3 destination signal (Fig. 3f). As reported before, the light-dependent activity of Arch is visible in the lysosome once v-ATPase is inhibited by Bafilomycin A1 (Baf)[30]. After 20 minutes of incubation with Baf (300 nM) and DPA (4 μM), light pulses (10 mW s² μm²; 560 nm; delivered at 0.3 Hz) induced the activation of Arch, causing lysosomal acidification as monitored by pHluorin (Fig. 3g). This acidification correlates well with a simultaneous increase in the fluorescence signal of the otherwise voltage-insensitive mKate protein (Fig. 3g, h). This indicates a redistribution of DPA molecules towards the luminal leaflet of the lysosomal membrane—away from mKate—, suggesting repolarization of the lysosomal membrane. Taken together, our results

indicate that hVoS$_{org}$ not only demonstrates the ability to follow the amplitude and kinetics of $\Delta\Psi_{ly}$ but also supports the importance of luminal pH in setting the resting potential of lysosomes.

**Calibration of the hVoS$_{org}$ signal.** In contrast to electrophysiological methods, the quantitative determination of the membrane potential with fluorescent dyes requires calibration. Aiming to obtain absolute values of resting $\Psi_{org}$ from optical measurements, we devised a protocol where the first step was an in cell calibration of the voltage-dependent quenching of Lamp1-EGFP. In a second step, the calibrated voltage versus fluorescence relationship was used to quantify the change in fractional fluorescence ($\Delta F/F$) corresponding to the transit between a condition of unknown lysosomal membrane potential (i.e., resting $\Psi_{ly}$ in the intact cell) to a condition where $\Psi_{ly}$ is known (i.e., 0 mV; obtained by permeabilization or rupture of the lysosomal membrane).

In cell calibration was done by adapting the FPP procedure aiming to control organellar membrane potential with potassium-based solutions (Fig. 4a). To this end, cells we first permeabilized with digitonin, leaving the lysosomal membrane intact and accessible. Following permeabilization, the cells were incubated with nigericin, an $H^+/K^+$ antiporter (to dissipate $\Delta pH_{ly}$), and valinomycin, a highly $K^+$-selective ionophore[34,35]. This combination makes potassium the main permeating ion at the lysosomal membrane and the membrane potential can be calculated according to the Nernst equilibrium for potassium[35,36]. Therefore, we obtained control over $\Psi_{ly}$ by simply changing the potassium concentration in the extracellular media (Fig. 4a). The average change in fractional fluorescence ($\Delta F/F$) was obtained at different concentrations of external potassium ($K_{out}$), under two conditions of luminal potassium ($K_{in}$). Low and high $K_{in}$ (1 mM and 130 mM, respectively) were obtained via 15 min equilibration procedure (Fig. 4a). In support of our experimental strategy, changes in $\Delta F/F$ are in good agreement with the prediction of Goldman-Hodgkin-Katz (GHK) flux equation for a single monovalent cation conductance[36,37]. The curve with high $K_{in}$ (positive range) has a permeability = 0.96 (Fig. 4b). In contrast, low $K_{in}$ curve (negative potentials) has a permeability = 0.72 (Fig. 4c). Thus, we obtained a relationship between $\Delta F/F$ and lysosomal membrane voltage (Fig. 4d; Supplementary Fig. 2a; 92 independent recordings; 570 lysosomes). Following a convention for single membrane organelles in the intact cell (i.e., lysosomes, endosomes, peroxisomes, ER, and Golgi), the membrane potential was calculated from $V_{cytosol} - V_{lumen}$[38].

As reported before for plasma membrane hVoS[17], we observed that the voltage dependence of hVoS$_{org}$ was non-linear and rectifying at positive and negative membrane potentials (Fig. 4d; Supplementary Fig. 2c). Our calibration curve showed a broad region from +120 mV to −30 mV (positive inside) where FRET signal is well described by a Boltzmann sigmoidal function[17–19,21,22], exhibiting a $V_{1/2}$ of +45 mV and steep voltage dependence of ~3% per 10 mV between +110 mV and −20 mV (Fig. 4d; Supplementary Fig. 2c). In addition, a linear response was also observed between −10 mV to −100 mV where hVoS$_{org}$ is weakly voltage-sensitive (~0.3% per 10 mV; $R^2 = 0.99$) (Fig. 4d). The calculated signal-to-noise ratio ($\mu_{sig}/\sigma_{sig}$) was 3.4 ± 1.2% $\Delta F/F$ (Supplementary Fig. 2b), corresponding to ~10 mV at positive membrane potentials and ~100 mV at negative membrane potentials. Overall, the half-maximal quenching voltage, voltage dependence, and S/N ratio are in agreement with all previous reports[18,19,21,22], suggesting that the general theory supporting hVoS FRET is robust enough to be extrapolated to any cellular membrane. Moreover, although a qualitative description of changes in $\Psi_{ly}$ can be performed within a wide range (i.e., from

−100 mV to +130 mV), quantification of $\Psi_{ly}$ is reliable within −10 mV to +110 mV (Supplementary Fig. 2c).

In a second step, we used the calculated $\Delta F/F$ versus voltage relation to obtain the absolute resting potential of lysosomes. To this end, we recorded $\Delta F/F$ transiting from rest (i.e., the fluorescence signal of lysosomes in the intact living cell incubated with DPA) to a condition where $\Psi_{ly}$ should be close to zero (i.e., digitonin-permeabilized lysosomal membrane) (Fig. 4e). In this case, ionophores were not used and digitonin remained in the solution until the end of the experiment (Fig. 4e; supplementary movie 1). The time course of these recordings showed stepwise changes in fluorescence that occur during incubations with digitonin (Fig. 4f, g). We reasoned that the first step corresponds to the breakdown of plasma membrane causing a change in the cytoplasmatic ionic concentration and that the second step corresponds to the permeabilization of the lysosomal membrane, making $\Psi_{ly} = 0$ mV. To test this hypothesis, we changed the ionic conditions by replacing extracellular $K^+$ with $Na^+$ ions. Although the initial step of fluorescence decay was abolished in a sodium-based solution, the absolute change in fractional fluorescence at equilibrium—representing the change in membrane voltage from resting to zero mV—remains fairly unchanged (Fig. 4h). Thus, would be reasonable to suggest that these intermediate steps are related to specific conductances that are native to the organelle type. Unveiling the contributions of these putative conductances is beyond the scope of the present work, therefore we did not explore this further.

By following this procedure, we estimated that the resting potential of the lysosome in HEK293 cells is about 56 ± 7 mV (lumen positive; $n = 7$; 106 regions of interest (ROIs)) (Fig. 4i). When observed in more detail, we noted that Lamp1-positive structures on the periphery have a smaller $\Psi_{ly}$ at rest (Supplementary Fig. 3). From the rim of the cell towards the perinuclear region, $\Psi_{ly}$ ranges from 30 to 60 mV and would not be unreasonable to propose that these differences can be explained by the pH gradient observed in lysosomes during maturation[39].

The distance from the FP donor to the midplane of the plasma membrane is a critical determinant of FRET efficiency in hVoS[19]. The Ro for the FRET pair DPA/EGFP is 37 Å and the optimal working distance of the donor FP to the midplane of the plasma membrane has been estimated between 40 and 70 Å[19]. Our Lamp1-EGFP was designed with a 26 amino-acid long linker between the membrane anchor and the FP, corresponding to ~30 to 45 Å according to previous estimates[21] or assuming a random coil[40]. With this in mind, we designed additional cytoplasm-facing reporters for Golgi (i.e., manosidase II fused to EGFP; EGFP-ManIIa) and ER (i.e., Sec61b fused to EGFP; Sec61b-EGFP) with engineered linkers of similar length (38 and 56 amino acids, respectively). By transiently transfecting HEK293 cells with either EGFP-ManIIa or Sec61b-EGFP we first confirmed that the voltage versus fractional fluorescence relationship for these new reporters was similar to the one observed for Lamp1-EGFP (Supplementary Fig. 2b).

Measuring the transmembrane potential of trans-Golgi network has been elusive and previously estimated close to 0 mV[41]. By transiently transfecting HEK293 cells with the Golgi marker EGFP-ManIIa we calculated a resting potential of 38 ± 6 mV (positive inside; $n = 4$; 55 ROIs) (Fig. 4i). On the other hand, previous calculations and measurements estimated ER's resting potential to be negative inside and somewhere between −100 mV and 0 mV[42–45]. In our hands, Sec61b-EGFP reported a resting membrane potential of −108 ± 12 mV (negative lumen; $n = 3$; 15 ROIs) (Fig. 4i). In this case, although the polarity (i.e., negative inside) can be undoubtedly established because of the positive fluorescence deflection observed, the intrinsic error of the

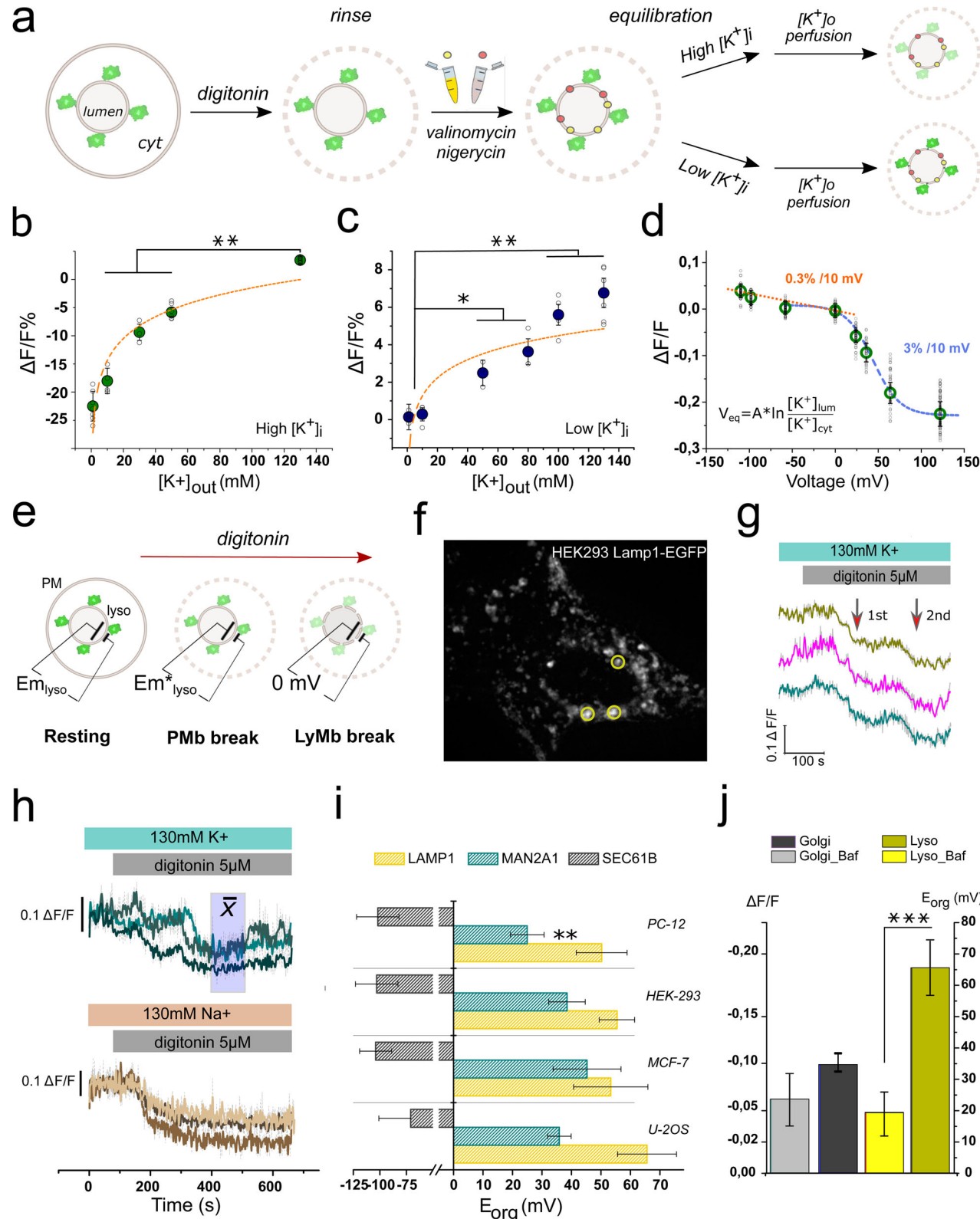

technique at negative potentials is large enough to stop us from reporting a reliable value for $\Psi_{ER}$ at rest (Fig. 4i).

Aware that cellular heterogeneity provides insight into cellular function, we used the same experimental approach to examine the resting transmembrane potential of lysosomes ($\Psi_{ly}$), Golgi ($\Psi_{go}$), and ER ($\Psi_{ER}$) endomembranes in different cell types. Four

epithelial-derived cell lines of mammalian origin were used (HEK293, MCF7, U2OS, PC-12). Although HEK293 cells are adenovirus-immortalized kidney cells, the others were originally obtained from naturally occurring cancer cells. PC-12 cells were obtained from a pheochromocytoma of the adrenal gland[46], MCF7 cells were isolated from breast cancer pleural effusion[47],

**Fig. 4 Organellar membrane potentials measured at rest. a** Schematic representation of the In-cell calibration protocol by potassium clamp. After gentle permeabilization with digitonin cells were incubated with reaching endomembranes. After equilibration with either low or high potassium solutions organelles were perfused with a set of potassium-based solutions. **b** Averaged ΔF/F values evoked by the different potassium solutions in high intraluminal potassium (green circles). Empty circles correspond to the average of independent experiments. Orange dashed line corresponds to the best fit to GHK equation. *$P < 0.05$; **$P < 0.01$. **c** Averaged ΔF/F values for the different potassium solutions in low intraluminal potassium (blue circles). Empty circles correspond to the average of independent experiments. Solid orange trace corresponds to the best fit to GHK equation. *$P < 0.05$; **$P < 0.01$. **d** ΔF/F versus membrane potential relationship ($n = 92$ independent experiments). Empty circles correspond to the full set of replicates. The data set was extracted from experiments in **b** and **c**. The blue trace corresponds to the best fit to a single Boltzmann distribution function. The orange line corresponds to a linear fit. **e** Schematic representation of the experiment designed to obtain the resting potential of intracellular membranes. Two sequential steps of permeabilization corresponding to the plasma membrane and the lysosomal membrane are expected. The second permeabilization step brings $\Psi_{ly}$ to zero before the final disruption of the lysosomal membrane. **f** Representative image of the cells used for calibration procedures. Three individual lysosomes are indicated in circles. **g** Representative traces used to estimate the resting potential. Cells were incubated in potassium-based extracellular ringer and exposed to 10 μM digitonin as indicated on top. Two steps of fluorescence quenching are indicated with red arrows. **h** Sodium on the external solution eliminates the intermediate quenching step without affecting the final equilibrium condition. Squares in blue shades indicate regions of the recording interpreted as the lysosomal membrane at 0 mV. Average values for the change in fractional fluorescence were extracted from the indicated region. **i** Calculated resting membrane potential of lysosomes ($n = 21$ independent experiments), Golgi ($n = 17$ independent experiments), and ER ($n = 14$ independent experiments) for the cell types indicated. Error bars indicate mean ± SD. **j** Effect of pH on the resting membrane potential of lysosomes and Golgi ($n = 6$ independent experiments). Error bars indicate mean ± SD; ***$P < 0.001$.

and U2OS cells derive from a moderately differentiated sarcoma of the tibia[48].

While the resting potential in lysosomes was similar for the cell lines surveyed, the resting potential of Golgi showed significant differences. The larger positive membrane potential observed in HEK293 (36 ± 1 mV; $n = 4$; 25 ROIs), U2OS (36 ± 3 mV; $n = 5$; 33 ROIs) and especially in MCF7 cells (46 ± 8 mV; $n = 4$; 38 ROIs) contrasts to the more depolarized potential observed in PC-12 (23 ± 3 mV; $n = 4$; 42 ROIs; $p = 0.004$ for MCF7 versus PC-12) (Fig. 4i).

We then compared the contribution of the proton gradient, $\Delta\mu H^+$, to the resting potential in lysosomes and Golgi. To this end, we incubated U2OS cells with Baf (300 nM) and recalculate $\Psi_{org}$ at rest in a condition where the proton gradient is dissipated. Although significant differences were observed on lysosomes ($p < 0.001$), non-significant differences were observed in Golgi (Fig. 4j). This result confirms the importance of $\Delta\mu H^+$ in settling the resting potential of lysosomes[49,50] and suggests a marginal contribution of $\Delta\mu H^+$ to the resting membrane potential of the Golgi membrane.

## Modulation of lysosomal membrane potential.

The mammalian target of rapamycin (mTOR) is a kinase that integrates intracellular level of nutrients, the energetic state, and growth factor signaling in higher eukaryotes[51]. It has been shown that starvation or treatments with mTOR inhibitors (e.g., rapamycin) induce a robust electrical response of the endosome/lysosome vesicular compartment, linking mTOR signaling and the activity of TPC sodium channels[52]. Moreover, the activity of mTORC1 has been associated to the activity of the SLC sodium-coupled amino-acid transporter and also to the activity of the lysosomal v-ATPase[53,54]. Consistent with the notion that the mTOR-signaling network is associated to lysosomal electrical activity, we observed that rapamycin (5 μM) evoked a strong and transient depolarization in intact lysosomes of living cells (Fig. 5a, b; Supplementary movie 2). The rapamycin-induced depolarization of the lysosomal membrane was observed in all cell types tested (Fig. 5c). Moreover, starvation (i.e., absence of amino acids and glucose in the media) builds up a larger resting potential (95 ± 22 mV; $n = 7$) and evokes a significantly larger lysosomal depolarization in response to rapamycin ($p < 0.01$) (Fig. 5d).

To explore the contribution of native lysosomal channels to the starvation response, the cells were transiently expressing either hTPC1 or hTRPML1 ion channels, deprived from nutrients for 1 h, and $\Psi_{ly}$ was calculated at rest. In both cases a more depolarized

resting potential was observed although TPC1 seems to be slightly more effective on collapsing the resting potential of the lysosome (18 ± 3 mV; $n = 5$; $p < 0.01$) than TRPML1 (25 ± 5 mV; $n = 5$; $p < 0.01$) (Fig. 5e). This suggests that overexpressed channels are active during starvation and that their activity cannot be compensated efficiently by either the v-ATPase or other control mechanisms operating at the lysosomal membrane (e.g., leak potassium channels). We then compared the averaged response of lysosomes to rapamycin in cells expressing Lamp1-EGFP alone or co-expressing either TPC1 or TRPML1 (Fig. 5f, g). We calculated that the rapamycin-dependent depolarization dissipates the lysosomal resting potential by 42 ± 4 mV ($n = 5$; 76 ROIs), which corresponds to ~75% reduction of $\Psi_{ly}$ at rest (Fig. 5g). In contrast, cells overexpressing either TPC1 (20 ± 6 mV; $n = 4$; 50 ROIs) or TRPML1 (12 ± 4 mV; $n = 5$; 45 ROIs) channels showed modest depolarizations in response to rapamycin ($p < 0.01$). Interestingly, a late repolarization phase consistently appeared ~+10 mV in every condition, even in the presence of rapamycin in the extracellular media (Fig. 5f, black arrows). Such repolarization component would be consistent with the activation of voltage-dependent conductances, native to the lysosomal membrane, helping in the restoration of the resting potential by shunting depolarization[25,27]. Moreover, the more depolarized $\Psi_{ly}$ observed in cells overexpressing cation-permeable lysosomal channels can explain the increased excitability observed in lysosomal patch-clamp experiments[6].

Overall, the overexpression of cation-permeable lysosomal channels alters both the resting potential and the dynamic response of the lysosome during depolarization, without major effects on the repolarization process.

## Discussion

By adapting a hybrid-FRET voltage sensor, we report here the resting membrane potential of different organelle compartments in living cells. Membrane potential is a major regulator of transport across membranes. Therefore, organelle function—and by extension the metabolic state of the cell—is modulated by the voltage gradient across organelle's membrane. Lysosomal membranes contain several voltage-gated ion channels and transporters, suggesting that a fine-tuning of lysosomal function is modulated by the membrane potential[24]. Imaging $\Delta\Psi_{ly}$ have been successfully accomplished in the past by using a combination of potentiometric fluorescent dyes (i.e., oxonol derivatives) forming a FRET pair with fluorophores that are preferentially segregated to the lysosome membrane[15] and recently with improved

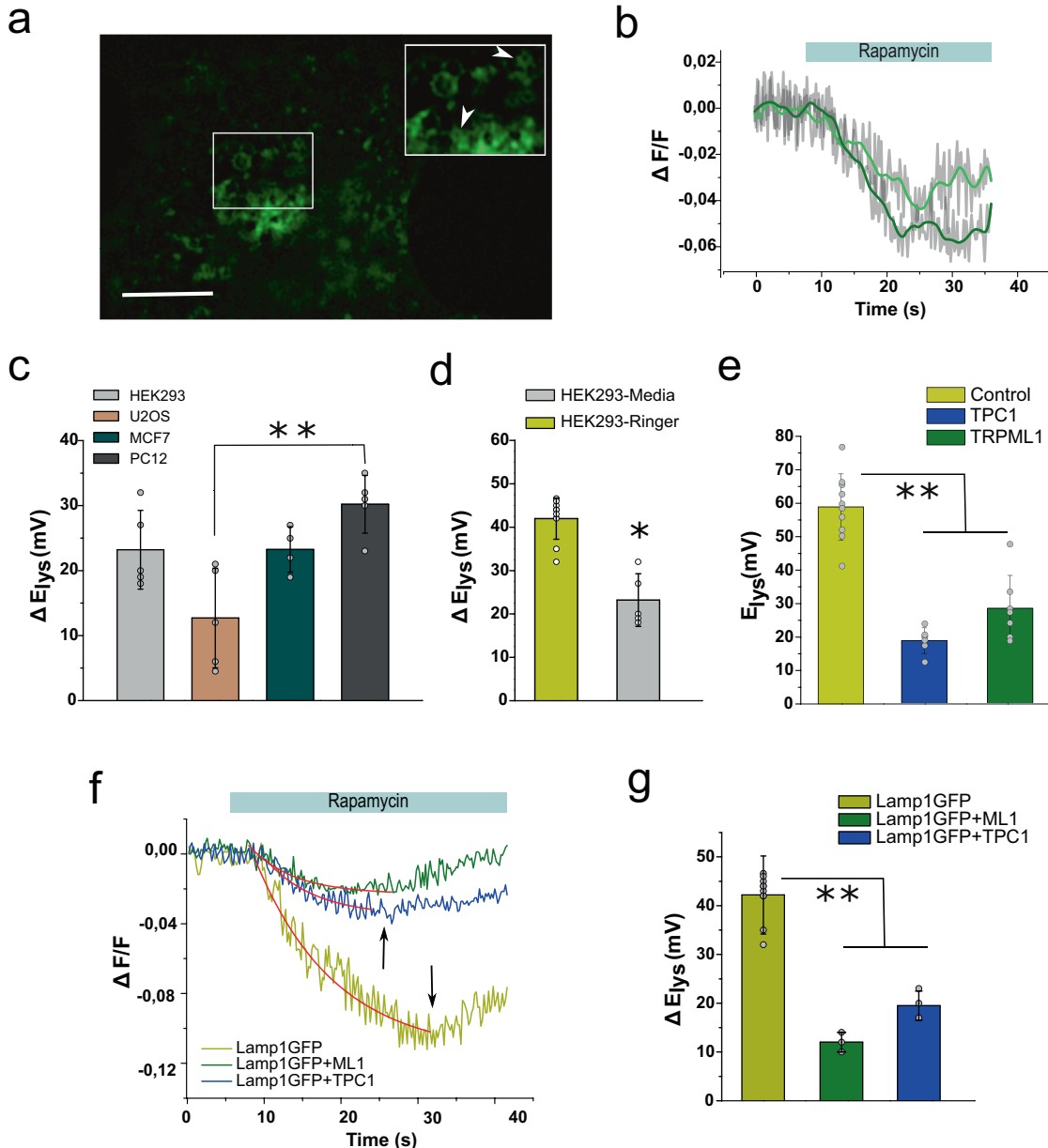

**Fig. 5 Time-resolved imaging of rapamycin-induced lysosomal depolarization. a** Representative image of HEK293 cells transiently expressing Lamp1-EGFP. Bar = 5 µm. Inset. Two individual lysosomes are indicated with arrowheads. **b** Changes in fractional fluorescence (ΔF/F) induced by rapamycin in the two lysosomes indicated in **a**. **c** Rapamycin-induced electrical response in the four different cell types indicated ($n = 7, 6, 5$, and 5 independent experiments for HEK, U2OS, PC-12, and MCF7 cells respectively). **d** A larger rapamycin-induced response is observed in cells that are deprived from nutrients ($n = 6$ independent experiments per condition). **e** Transient overexpression of lysosomal resident channels alters normal membrane potential ($n = 5$ independent experiments per condition). **f** Representative traces of rapamycin-induced depolarization of lysosomes overexpressing lysosomal resident channels. Red traces correspond to first-order exponential decay function. Arrows indicate when membrane potential reaches 10 mV. **g** Changes in the lysosomal membrane potential in the presence of the mTOR inhibitor rapamycin (5 µM) ($n = 5$ independent experiments). Error bars indicate mean±SD; **$P < 0.01$.

DNA-based probes targeting lysosomes and trans-Golgi network by taking advantage of trafficking routes[16]. However, the high density and variety of membrane-bound organelles make impossible to properly isolate the individual contribution of endosomes, lysosomes, or phagosomes when a FRET pair is formed by combining hydrophobic dyes or using trafficking routes to distribute the probes. In this work we have introduced hVoS$_{org}$, an experimental approach that can accurately detect variations in the membrane potential of organelles minimizing restrictions imposed by spatial resolution and probe targeting. The differences in targeting strategies of these different

approaches translate into more precise measurements for the case of hVoS$_{org}$ when compared to any other method developed so far. The dynamic range for cytoplasmic-facing probes described here is restricted to −20 to +100 mV. In addition, we showed that cytoplasmic-facing probes are largely inefficient to describe changes in membrane potential in the negative range (e.g., endoplasmic reticulum). Proper calibration of lumen-facing probes could be helpful in surmounting this issue. Moreover, although our results are in general agreement with recently reported values of lysosomal electrical activity obtained with DNA-based probes, the large membrane potential observed in

TGN contrast to the modest resting potential we observed in Golgi by using EGFP-ManIIa as reporter[16].

Overall, our results in lysosomal membranes are in good agreement with previous reports showing (i) a large contribution of the proton gradient to the resting potential in lysosomes[31,50]; (ii) a direct relationship between the lysosomal membrane potential and the metabolic state of the cell[52]; (iii) the existence of a voltage-dependent shunt controlling repolarization in intact lysosomes[27].

When a FRET system is set between a pair of light-emitting probes, spectral properties have to be controlled to avoid optical leak and bleed-through. As the FRET pair used here consists of a fluorescent protein donor and a non-fluorescent acceptor, bleed-through correction is no longer required. By providing a non-fluorescent FRET acceptor reaching all intracellular membranes in combination with extensively validated organelle markers, our single wavelength excitation method contributes on solving the problem of targeting to specific intracellular membranes together with providing a larger selection of wavelengths to be used.

## Conclusion

By targeting hVoS$_{org}$, we effectively measured changes in membrane potential of unitary organelles of different types, resolved in time and space within living cells. The present hVoS$_{org}$ approach demonstrated to be robust enough to measure the elusive resting potential of intact individual subcellular structures that include lysosomes and Golgi. The signal-to-noise ratio, the potential for optical multiplexing, and the compatibility with multi-photon excitation make hVoS$_{org}$ ideal to perform simultaneous measurements on intracellular membranes not only in vitro but also in living tissue.

## Methods

**Cell culture and clones**. HEK293 cells were cultured in Dulbecco's Modified Eagle Medium (DMEM) supplied with 10% fetal bovine serum (FBS). Cells were plated in poly-l-lysine coated coverslips and transfected using lipofectamine 2000 (Invitrogen). Recordings were made 24–36 hours after transfection. Lamp1GFP was a gift from Patricia Burgos (Universidad Austral de Chile). hTPC1 was a gift from Dejian Ren (University of Pennsylvania), TRPML1 was a gift from Kirill Kiselyov (University of Pittsburg). EGFP versions of hTPC1 (Addgene plasmid # 165981) and hTRPML1 (Addgene plasmid # 165982) were obtained by PCR amplification. Farsenylated EGFP was kindly provided by from Baron Chanda, hVoS (Addgene plasmid # 45282) was a gift from Meyer B. Jackson. Lyso-pHluorin (Addgene plasmid # 70113) and Lyso-pHoenix (Addgene plasmid # 70112) were a gift from Christian Rosenmund. EGFP-ManII (Addgene plasmid # 165979) was obtained by PCR amplification from sapphire-ManII and further subcloning in pEGFP-C1 vector. Sec61b-EGFP (Addgene plasmid # 165980) was obtained by PCR amplification from halo-Sec61b. Sapphire-manosidase II and halo-Sec61b were kindly provided by Jennifer Lippincott-Schwartz (HHMI, Janelia).

**Reagents**. Dipicrylamine sodium salt was obtained from Biotium Inc. (Fremont, CA). Rapamycin, dimethyl sulfoxide (DMSO), and ammonium chloride were obtained from Sigma-Aldrich. Valinomycin and nigericin were obtained from Tocris Bioscience (Bristol, UK). Standard salts used for solutions were obtained from Merck.

**Solutions and drug delivery**. Dipicrylamine sodium salt (DPA) was prepared fresh at 20 mM stocks in DMSO and used at a final concentration of 4 µM. Extracellular solution 1 contained NaCl (140 mM), KCl (5 mM), Ca Cl$_2$ (2 mM), Mg Cl$_2$ (2 mM), Glucose (5 mM), 4-(2-hydroxyethyl)-1-piperazineethanesulfonic acid (HEPES) (8 mM) at pH 7.34. Extracellular solution 2 consisted in DMEM/F12 supplemented with 10% FBS without phenol red. Cells were incubated with DPA for at least 25 minutes before starting the experiments. In all the experiments the ringer solution contained DPA (4 µM) and DMSO (0.15 % v/v). At all times the concentration of both DPA and DMSO remained constant in the external solution. The drug-delivering pipette was placed close to the cell using a mechanical manipulator (Narishige, Tokio, Japan) and was pressure-ejected using a microliter syringe. Gravity perfusion was digitally controlled by a custom-made arduino-based circuit.

*Calibration curve*. To make the voltage versus fluorescence calibration curve cells were first gently permeabilized with digitonin (10 µM, 1 min) in a solution containing KCl (130 mM), NaCl (10 mM), Ca Cl$_2$ (2 mM), MgCl$_2$ (2 mM), MgATP (5 mM), Glucose (5 mM), HEPES 8 mM at pH 7.4. Nigericin and valinomycin were delivered to the extracellular solution (now in contact with the lysosomal membrane) after rinsing the detergent out of the recording chamber. Before exchanging solutions, lysosomes were equilibrated with the corresponding K$^+$ buffer (low or high concentration) for 15 mins, in the presence of Nigericin and valinomycin.

### Image acquisition

*Voltage imaging*. An Orca Flash 4.0 CMOS camera (Hamamatsu Photonics, Japan) mounted on an Olympus IX71 microscope was used to image hVoS$_{org}$ fluorescence. Images were taken under normal epifluorescence, using a water immersion objective (×60, N.A. = 1.3). A 473 nm (Melles Griot, Carlsbad, CA) and 532 nm (LaserGlow, Toronto, Canada) diode pump lasers were transmitted via the rear illumination port of the microscope and reflected the sample by a double dichroic mirror with reflection bands at 473–490 nm and 530–534 nm in combination with filters having transmission bands at 500–518 nm and 550–613 nm (Semrock, Rochester, NY). Excitation was controlled by a mechanical shutter (Uniblitz, VA Inc., Rochester, NY). The acquisition was performed on individual cells at 20–10 Hz without binning. Image acquisition was controlled by micro-manager (Open Imaging, San Francisco, CA).

*Space-correlated imaging*. Colocalization of organellar markers was done by imaging with a Leica DMi8 Confocal Laser Scanning Microscope. Fluorescence images were captured at 20 °C by an EMCCD camera (iXon888, Andor Instruments) using a ×60 objective (N.A. = 1.40; water) and suitable filter sets for EGFP and mCherry (ex. 488 and 561, respectively).

*Optogenetics*. To image lyso-phoenix signals and to optically activate the proton pump Arch, we used an inverted Zeiss LSM 880 microscope equipped with an Airyscan array detector. A ×63, 1.4 NA, oil-immersion objective was used. Imaging was performed at 20°C and 5% CO$_2$. Signals coming from pHluorin and mKate were imaged at 5 Hz (50-ms exposure per channel) using 488 and 595 nm laser lines, respectively. The fluorescence signal was collected within 510–540 nm for the case of pHluorin and within 630 and 670 for the case of mKate. The pump activity was induced by light pulses delivered at 0.3 Hz (10 mW s$^2$ µm$^2$) during the recording. Acquisition, visualization, and Airyscan postprocessing were performed using Zen software (Zeiss).

### Signal analysis

*Colocalization*. After background subtraction, 20 frames of space-correlated images were averaged and overlay images were produced (https://imagej.nih.gov/ij/). To evaluate pixel colocalization we calculate Pearson's coefficient using JACoP plugin for ImageJ[55].

*Mobility maps*. We performed the calculation following the protocol in Brauchi et al.[28]. The fluorescence signal was lower threshold over two standard deviations above the mean of the camera noise. The upper threshold was set to identify the desired lysosomal-shaped objects. Threshold images were converted into binary format events. The mobility function was calculated for each pixel of the image sequence.

*Fluctuation of fluorescence*. To calculate fluorescence kinetics on pHluorin and DPA/EGFP experiments, ROIs were established on doughnut-shaped, Lamp1-positive spots having a size between 200 and 1500 nm. Several ROIs per cell were selected from background-subtracted stacks. All together shape, size, mobility, and localization helped us to define the ROI set that was measured on each cell. After selection, the fluorescence time course was recorded for each ROI. Changes in fractional fluorescence were calculated according to: $\Delta F/F = ((F_n - F_0)/F_0)$, where $F_n$ is the corrected fluorescence at frame $n$, and $F_0$ corresponds to the average of at least 10 frames of the baseline fluorescence. Baseline fluorescence corresponds to the steady-state GFP signal after DPA equilibration. Once $\Delta F/F$ was calculated bleaching was corrected by using a single exponential function fitting the decay of fluorescence at baseline. Data were filtered by an FFT filter with a cutoff one-fourth of the sampling frequency.

*Calibration curve*. Lysosomal membrane voltage ($V_{ly}$) for the different potassium solutions was calculated by using the Nernst equation.

$$V_{ly} = \frac{RT}{zF}\ln\left(\frac{[K^+]_{lumen}}{[K^+]_{external}}\right) \qquad (1)$$

where $z$ corresponds to ion charge; $F$, $R$, and $T$ have their usual meanings.

Fractional fluorescence was plotted against the external concentration of potassium and that relation was fitted to the GHK flux equation[37], assuming a direct relationship between ion flux through ionomycin (creating a change in membrane polarity) and the fluorescence quenching reporting changes in

membrane polarity.

$$\frac{\Delta F}{F} = y_0 + \left(pz^2 FV_{ly}\right)[K^+]_{ext}\left(\frac{e^{z\left(V_{ly} - V_n\right)} - 1}{e^{zV_{ly}} - 1}\right) \quad (2)$$

where $V_n$ is the Nernst potential at equilibrium and $p$ corresponds to permeability.

The fractional fluorescence versus voltage relation was fitted to a linear function at negative potentials. From $-30$ mV to $+140$ mV, the relationship was fitted to a single Boltzmann distribution function.

$$\frac{\Delta F}{F} = \frac{\frac{\Delta F}{F_{max}}}{1 + e^{-\left[\frac{zF\left(V - V_{0.5}\right)}{RT}\right]}} \quad (3)$$

where $z = 1$, $V_{0.5}$ is the half-activation voltage, and $\Delta F/F_{max}$ was calculated from the steady-state average.

**Mathematical Model for DPA distribution.** A multi-compartment model consisting of an extracellular reservoir (E), extracellular (M1), and intracellular membrane leaflets (M2), and a cytosolic compartment (C) was implemented as a series of differential equations in Python language. The rate constant from extracellular and intracellular compartments to the membrane was the same (a), and rates from the membrane to both extracellular and intracellular compartments were the same (b)[56]. Rates between the two membrane leaflets (i.e., $v1$ and $v2$) were obtained by extracting time constants for charge movement from whole-cell membrane capacitance measurements in presence of 4 μM DPA in mast cells[57]. Time constants followed a bell-shaped curve as a function of voltage. We fitted the percentage of quenching from negative voltages to the peak according to:

$$v_2 = B_0 e^{\left(\frac{-slp_2 VF}{RT}\right)} \quad (4)$$

On the other hand, the reverse equilibrium was fitted to:

$$v_1 = A_0 e^{\left(\frac{-slp_1 VF}{RT}\right)} \quad (5)$$

Overall, we obtained $A0$, $B0$, slp1, and slp2, which were used for the final model of distribution. For all accounts, $R$ is the gas constant, $T$ is the absolute temperature and, F corresponds to the Faraday constant; $V$ was assumed $-30$ mV for the plasma membrane and $+60$ for the lysosomal membrane. Fluorescence quenching was fitted to the model in order to obtain the rate constants $a$ and $b$, parameters $A0$, $B0$, slp1, and slp2 and to calculate the rate constants $v1$ and $v2$. The model calculates probabilities to find molecules in each compartment. Therefore, by assuming DPA density at equilibrium $10^{-4}$ molecules/A$^2$ [57], a cell with diameter of 40 μm and a membrane width of 10 nm, we were able to transform probabilities to concentration at the membrane and cytosol. To calculate the equilibrium concentration of DPA at the lumen of the lysosome we considered a lipidic sphere with a diameter of 1000 nm and the external concentration of cytosolic DPA at equilibrium (3 μM).

**Statistics and figure preparation.** Voltage measurements. Individual cells on one day of transfection contribute with several lysosomes (in the order of 10–25 per cell), which are considered as replicates. Therefore, the average value from organelles within a group of cells coming from the same transfection corresponds to $n = 1$ and contains the information from several individual lysosomes. All values presented correspond to the mean and standard deviation of the averaged signals. The final statistical procedure was done by performing a one-way analysis of variance comparing the differences observed in several independent experiments per condition. A $p$ value of 0.05 was considered significant and a Bonferroni post-test was performed for all pairwise comparisons tests. Normality was evaluated by Kolmogorov–Smirnov test. All numerical data was treated, and the statistical analysis was computed using Microcal OriginPro ver9 (OriginLab corporation, Wellesley Hills, MA). Figures were prepared using Microcal OriginPro ver9, ImageJ, Biorender, and Inkscape.

**Reporting summary.** Further information on research design is available in the Nature Research Reporting Summary linked to this article.

## Data availability
The data sets generated and analyzed during the current study are available in the Dryad repository, https://doi.org/10.5061/dryad.n02v6www1. Any remaining data are available from the authors upon request. Fluorescent protein constructs are available from AddGene.

## Code availability
The codes generated and used during the current study are available in the GitHub repository, https://github.com/brauchilab/DPA-quenching-model.

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

## Acknowledgements
We thank David E Clapham (HHMI) and the Clapham group for all the technical support and advice; Kirill Kiselyov (U Pittsburg) for his helpful insights; Alexandria Miller (HHMI) for her critical reading of the manuscript. This work was supported by Anillo Cientifico #ACT1401, FONDECYT 1191868, and ANID-Millennium Science Initiative Program #NC160011. S.B. is part of UACh Program for Cell Biology & CISNe. P.A.R. is funded by DICYT-USACH. E.M. is a CONICYT fellow. J.P.V. is now at the Department of Science, Santiago College, Chile.

## Author contributions
E.M., J.P.V., and S.E.B. performed experiments including cell culture, molecular biology, and imaging. P.A.R. performed simulations. S.E.B. and E.M. designed experiments and analyzed data. C.C. built and programmed the perfusion system and digital control. E.M. and S.E.B. prepared the figures. E.M., P.A.R., and S.E.B. wrote the manuscript.

## Competing interests
The authors declare no competing interests.
