## [Peer Review File · Communications Biology]

Reviewers' comments:

Reviewer #2 (Remarks to the Author):

In this paper by Matamala and Brauchi et al. the voltage-sensitive FRET quencher DPA and an organelle targeted fluorescent protein is developed and used to map voltage across internal membrane compartments.

This paper is an interesting combination of methods development and application. It is an important and relatively unexplored area of cellular biophysics and I find the underlying method interesting and quite simple to implement. Thus, it appears there could be a relatively easy adoption of this method by the membrane biophysics/organelle cell biology communities. I do, however, have several reservations about the paper that should be addressed before publication. I will list my specific concerns below.

1. My primary issue with the paper is that after multiple readings I am still not convinced that the appropriate calibration measurements were done to determine the absolute resting voltage values and changes in voltage in the different organelles studied (Figure 3). First, the absolute quenching values and delta quenching values for DPA should be dependent on the final concentration of DPA and the combined distance of these DPA FRET quenchers to the donor fluorophore. As each organelle is tagged with a FP at a different location relative to the target membrane, and each membrane might have different concentrations of DPA, it is not clear 1) what the absolute quenching should be and 2) what the slope of the response should be given the concentrations of DPA and the distances involved. Thus, it appears difficult to make these calibrations with certainty. The authors should at least present a computational model to help the reader understand how these variables would affect the voltage-dependence fluorescence response in each target organelle. As it stands now, I do not understand how the authors are sure that their measurements can be reported in mV. This should be more clearly addressed in the paper. As a tool paper this is a key addition to the work.

2. The figures are quite small and difficult to evaluate at their current size and presentation. This is true for all graphs and images in the paper. It is almost impossible to evaluate figures 1 and 2 in their current format.

3. As far as I can tell all experiments were done in one cell type (HEK293). Do the authors know if the system works in other mammalian cell types? It would be particularly interesting in electrically active tissues such as muscle, endocrine cells, or neurons. Likewise, could the system be used in tissues? This should be discussed in the revised manuscript.

4. There are several typos.

"at the level of single organelle."

"sensor composed by"

"by doing this, we observed a linear of"

Reviewer #3 (Remarks to the Author):

Notes on Matamala et al

I have read "Imaging the electrical activity of organelles in living cells" by Matamala et al. In this

manuscript, the authors propose a novel method for quantifying intracellular membrane potentials by combining organelle-targeted fluorescent proteins and the dipicylamine which whose partitioning is depends upon the transmembrane potential. The method is relatively simple and should provide a powerful tool for measuring intracellular membrane potentials. Overall, the publication is important and should be of broad interest to the researchers of Communications Biology. However, prior to publication, the manuscript needs significant revision to improve the rational and clarity. The some of the concepts are new, and clarity will be important for reaching a broad audience. I have made a number of specific items below.

Major concern – My overriding concern is that the theory of underpinning the approach is underdeveloped. The authors ascribe the observed quenching of fluorescent proteins as FRET with DPA. Some simple calculations regarding the expected Forster distances and necessary concentrations would be helpful in making this case. Especially given that the absorbance spectrum of DPA would suggest a differential sensitivity for the FPs used (e.g. GFP and mCherry). A table providing the Forster distances, should be provided for each FP used. An illustration of the overlapping donor and acceptor spectra would be very helpful. Lastly, a prediction of the concentration dependence of FP quenching/FRET with DPA alongside a dose response, would provide a clear theoretical basis for the optical effects underpinning the method.

Specific concerns:

Figure 1, Dipicylamine is misspelled.

The example in Fig 1d top row appears to show extensive LAMP1 association with the plasma membrane. This localization does not seem to occur in the other examples of LAMP1 shown. Was this image truly representative?

Figure 2 seems to be missing labels. Presumably the columns in Fig2a reflect different points in time? The topology argument is a little hard to follow given that the diagram depicts GFP in the lumen of the pHoenix construct, but in the cytosol for the LAMP construct.

For Fig 2d, a cleaner result might be afforded by using the VATPase inhibitor the quenching and cytosolic alkalization created by NH₄Cl...

Fig3b, it is not clear why digitonin and ionophores were used together. Presumably the goal was to do permeabilize only the plasma membrane to allow the ionophores accelerated access to the lysosome? The results text should be improved to clearly articulate

Fig4 – placing labels next to the columns would make the figure easier to read.

237. As the FRET pair used here consists of a fluorescent protein donor

238. 249 and a colorless acceptor,

239. by changing the paradigm,

240. 250 providing a colorless FRET acceptor reaching intracellular

241.

Please note that the acceptor is not 'colorless,' if it were, FRET would not be the mechanism. I think you mean non-fluorescent.

We appreciate the reviewers' comments on our study. Their constructive critiques were very helpful in improving the manuscript. Major changes in the revised manuscript including an extended calibration curve, measurements in different cell types, and a mathematical model to describe the distribution of the voltage sensor in the different cellular compartments. Additionally, figures were improved and the text now displays a revision of wording throughout the entire manuscript. Changes directly related to reviewers' comments are discussed below.

Point by point response to the referees

Reviewers' comments:

Reviewer #2 (Remarks to the Author):

In this paper by Matamala and Brauchi et al. the voltage-sensitive FRET quencher DPA and an organelle targeted fluorescent protein is developed and used to map voltage across internal membrane compartments. This paper is an interesting combination of methods development and application. It is an important and relatively unexplored area of cellular biophysics and I find the underlying method interesting and quite simple to implement. Thus, it appears there could be a relatively easy adoption of this method by the membrane biophysics/organelle cell biology communities.

We thank the positive feedback.

I do, however, have several reservations about the paper that should be addressed before publication. I will list my specific concerns below.

1. My primary issue with the paper is that after multiple readings I am still not convinced that the appropriate calibration measurements were done to determine the absolute resting voltage values and changes in voltage in the different organelles studied (Figure 3). *First, the absolute quenching values and delta quenching values for DPA should be dependent on the final concentration of DPA and the combined distance of these DPA FRET quenchers to the donor fluorophore.* As each organelle is tagged with a FP at a different location relative to the target membrane, and each membrane might have different concentrations of DPA, it is not clear what the absolute quenching should be and what the slope of the response should be given the concentrations of DPA and the distances involved. Thus, it appears difficult to make these calibrations with certainty.

This is correct, DPA concentration and XFP distance to the membrane will determine the magnitude of the FRET signal.

However, DPA reaches saturation at the membrane. Our model of DPA distribution considered previous calculations and estimates that such

equilibrium is reached after about 25 minutes of DPA incubation. All our incubation protocols were adjusted accordingly. The observed quenching of XFPs does not correspond to DPA enriching or exiting the membrane it corresponds to changes in redistribution of the voltage sensitive particle (DPA) between membrane leaflets. Our model and previous observations predicts that DPA exit from the membrane is not favorable, therefore the concentration at the membrane is not expected to change during the course of the experiments (See annex1 and reference 21).

On the other hand, the distance between the elements of the FRET pair is certainly critical. For that reason we performed again all the quantitative measurements using EGFP and a linker of 26 amino acids for the case of Lamp1. Such number of amino acids in the linker is the same used in at least two experimental articles dealing with the issue of improving the FRET efficiency for this particular pair (references 19 and 21).

Our calibration curves for the newly engineered Golgi and ER markers (Supplementary figure 2) are in agreement with all previous works, and our own calibration of Lamp1EGFP, demonstrating not only that the design of the experiment (i.e. in cell potassium voltage clamp) was adequate enough but also that the FRET pair itself is responding in similar fashion at different cellular membranes.

2. The authors should at least present an computational model to help the reader understand how these variables would affect the voltage-dependence fluorescence response in each target organelle. As it stands now, I do not understand how the authors are sure that their measurements can be reported in mV. This should be more clearly addressed in the paper. As a tool paper this is a key addition to the work.

We thank this critique that helped us to explain our method and calibration better.

A model, similar to the one suggested by the reviewer, was already published twice (References 19 and 21 in the manuscript body). To make our observations comparable with these works we repeated the whole set of experiments using markers with linkers of similar size. These linkers were engineered within the parameters considered in both, previous observations and mathematical models. Quantification was restricted to EGFP to make our measurements comparable between organelles. Nevertheless, we implemented an additional model in python that is now part of the manuscript (Appendix 1). The model allows for the calculation of DPA's distribution and concentration at membrane leaflets (i.e.

PlasmaMembrane out, PlasmaMembrane in; OrganelleMembrane out, OrganelleMembrane in), soluble fractions, and estimate/predict the time course of DPA-dependent quenching at rest.

Regarding the calibration method, we provide a better description in the revised text. Additionally we provide a schematic description of the procedure (Figure 4a). Briefly, we obtained the values in mV from the Nernst equilibrium for potassium that is established at the lysosomal membrane after we incubate the sample with selective ionophores. This is an adaptation of a well established procedure that now is well referenced in the text. To confirm the ionophores (specially valinomycin) is carrying most of the current in during the in cell calibration, we fitted a GHK equation and the permeability was calculated between 0.7 and 0.96 at negative and positive potentials respectively. This indicate that specially the most sensitive part of the curve (between -20 and 120 mV, see supplementary figure 2) is well calibrated and very reliable.

Because of the improvements in our experimental procedures and calibration curve we have corrected the values we reported in the first version of this manuscript. Less depolarized potentials were observed for the case of lysosomes (~35 mV less depolarized) and Golgi (~40 mV less depolarized) present significant differences compared to the previous version. This correction would have not been possible without reviewers critiques to the calibration procedure.

3. The figures are quite small and difficult to evaluate at their current size and presentation. This is true for all graphs and images in the paper. It is almost impossible to evaluate figures 1 and 2 in their current format.

We apologize for the inconvenience. Figures were reformatted to improve presentation of the data.

4. As far as I can tell all experiments were done in one cell type (HEK293). Do the authors know if the system works in other mammalian cell types? It would be particular interesting in electrically active tissues such as muscle, endocrine cells, or neurons. Likewise, could the system be used in tissues? This should be discussed in the revised manuscript.

Now we have incorporated data extracted from four mammalian cell types. This was done not only for the absolute values of resting potential (figure 4) but also for the lysosomal response to rapamycin (figure 5). We didn't try excitable cell types such as neurons or muscle cells. We don't have access to the later and for the case of neurons, the small volume of the cell body

prevent easy access to single organelle measurements which is the main point we tried to make here.

Nevertheless, we explored the effect of plasma membrane depolarization on the lysosomal quenching. A set of preliminar experiments suggests that some crosstalk between the two compartments might exist. However, we still don't have suitable control experiments and the dispersion of the data is large enough to prevent proper interpretation. Although we acknowledge that the experiment is of value, additional control experiments are needed to clarify this issue.

4. There are several typos.

“at the level of single organelle.”

“sensor composed by”

“by doing this, we observed a linear of”

These are now corrected.

Reviewer #3 (Remarks to the Author):

Notes on Matamala et al

I have read “Imaging the electrical activity of organelles in living cells” by Matamala et al. In this manuscript, the authors propose a novel method for quantifying intracellular membrane potentials by combining organelle-targeted fluorescent proteins and the dipicylamine which whose partitioning is depends upon the transmembrane potential. The method is relatively simple and should provide a powerful tool for measuring intracellular membrane potentials.

Overall, the publication is important and should be of broad interest to the researchers of Communications Biology.

We appreciate the positive feedback from the reviewer.

However, prior to publication, the manuscript needs significant revision to improve the rational and clarity. The some of the concepts are new, and clarity will be important for reaching a broad audience. I have made a number of specific items below.

Major concern – My overriding concern is that the theory of underpinning the approach is underdeveloped.

It was our mistake not to introduce this properly in the first version.

There is a large body of data and mathematical models describing the characteristics of this particular FRET pair. Thus, the present manuscript includes:

1) a better reference to previous works that have studied the FRET pair in detail and developed mathematical models to interpret their results (REFs

17 to 21). These references not only present calibrations and models but also highlights the adaptability of the technique that has been used in cell and tissue under single photon excitation but also under multi-photon illumination. 2) a mathematical model describing the distribution of DPA in the different cellular compartments. To implement this, we have prepared a python script that is now part of the manuscript (see annex 1). To make our observations comparable with these previous works we repeated the whole set of experiments using EGFP only, with linkers engineered according to the parameters considered in both, previous experimental observations, and mathematical models. Our calibration curve is in full agreement with the literature available. The significant improvements performed to the calibration curve rendered voltage membrane values that are about 30% smaller in amplitude when compared to the previous version.

The authors ascribe the observed quenching of fluorescent proteins as FRET with DPA. Some simple calculations regarding the expected Forster distances and necessary concentrations would be helpful in making this case.

This is now described in page 13 line 242.

“The distance from the FP donor to the midplane of the plasma membrane is a critical determinant of FRET efficiency in hVoS¹⁹. The R_0 for the FRET pair DPA/EGFP is 37 Å and the optimal working distance of the donor FP to the mid plane of the plasma membrane has been estimated between 40 and 70 Å¹⁹. Our Lamp1-EGFP was designed with a 26 amino acid long linker between the membrane anchor and the FP, corresponding to ~30 to 45 Å according to previous estimates²¹ or assuming a random coil⁴⁰. With this in mind, we designed additional cytoplasm-facing reporters for Golgi (i.e. mannosidase II fused to EGFP; EGFP-ManII) and ER (i.e. Sec61b fused to EGFP; Sec61b-EGFP) with engineered linkers of similar length (38 and 56 amino acids respectively). By transiently transfecting HEK293 cells with either EGFP-ManII or Sec61b-EGFP we first confirmed that the voltage versus fractional fluorescence relationship for these new reporters was similar to the one observed for Lamp1-EGFP (Supplementary Fig. 2b).”

Especially given that the absorbance spectrum of DPA would suggest a differential sensitivity for the FPs used (e.g. GFP and mCherry). A table providing the Forster distances, should be provided for each FP used.

The reviewers were right, without proper design of the markers any comparison was not straight forward. Thus, we decided to repeat all the quantitative measurements using EGFP-tagged constructs, paying attention to use linkers of similar size. This allowed us to compare the performance of the FRET pair at different membranes using different anchors.

An illustration of the overlapping donor and acceptor spectra would be very helpful.

This is now provided in figure 1.

Lastly, a prediction of the concentration dependence of FP quenching/FRET with DPA alongside a dose response, would provide a clear theoretical basis for the optical effects underpinning the method.

This is provided as part of annex 1 (modeling of DPA distribution over time).

Specific concerns:

Figure 1, Dipicrylamine is misspelled.

Fixed.

The example in Fig 1d top row appears to show extensive LAMP1 association with the plasma membrane. This localization does not seem to occur in the other examples of LAMP1 shown. Was this image truly representative?

We have now a better exemplary images on figure 2 and supplementary figure 1.

Figure 2 seems to be missing labels. Presumably the columns in Fig2a reflect different points in time?

The figure was redesigned to address the critique.

The topology argument is a little hard to follow given that the diagram depicts GFP in the lumen of the pHoenix construct, but in the cytosol for the LAMP construct.

The figure was redesigned to address the critique, we present a schematic representation accompanying the results.

For Fig 2d, a cleaner result might be afforded by using the VATPase inhibitor the quenching and cytosolic alkalization created by NH₄Cl...

We explored this recommendation by measuring the effect of VATPase inhibition to the membrane potential of lysosomes and golgi. We observed that there is a good correlation between the lumen acidity and changes in membrane voltage (figure 4 f). Moreover, it was encouraging to observe that in Golgi, where low pH is seemingly not a requirement for organellar function, the effect of bafilomicyn 2A is not significant.

Fig3b, it is not clear why digitonin and ionophores were used together. Presumably the goal was to do permeabilize only the plasma membrane to allow the ionophores accelerated access to the lysosome? The results text should be improved to clearly articulate

The text was improved and aiming better description of our protocols, schematic representations for the experimental procedures are provided. The ionophores are used for the purpose of 1) dissipate the contribution of the pH gradient and 2) provide a selective pore for potassium so the potassium clamp can operate. The good fit to the GHK equation (permeability > 0.95 at positive potentials) supports our experimental design and suggest a good incorporation of the ionophores at the lysosomal membrane.

Fig4 – placing labels next to the columns would make the figure easier to read. **All figures were redesigned to improve the presentation of the data and readability.**

237. As the FRET pair used here consists of a fluorescent protein donor

238. 249 and a colorless acceptor,

239. y changing the paradigm,

240. 250 providing a colorless FRET acceptor reaching intracellular

241. Please note that the acceptor is not 'colorless,' if it were, FRET would not be the mechanism. I think you mean non-fluorescent.

The reviewer is correct. This misconception was corrected throughout the manuscript.

REVIEWERS' COMMENTS:

Reviewer #2 (Remarks to the Author):

The authors have addressed all the major concerns from my first review. The paper is much improved.

Reviewer #3 (Remarks to the Author):

The authors have addressed all of my major concerns. This is a very nice body of work that should be of broad interest to the scientific community.

I recommend publication in Communications Biology.